# Molecular Biomarkers of Oxygen Therapy in Patients with Diabetic Foot Ulcers

**DOI:** 10.3390/biom11070925

**Published:** 2021-06-22

**Authors:** Alisha R. Oropallo, Thomas E. Serena, David G. Armstrong, Mark Q. Niederauer

**Affiliations:** 1Comprehensive Wound Healing Center and Hyperbarics, Department of Vascular Surgery, Zucker School of Medicine Hofstra/Northwell, Hempstead, NY 11549, USA; 2Serena Group Research Foundation, Cambridge, MA 02140, USA; serena@serenagroups.com; 3Limb Preservation Program, Department of Surgery, Keck School of Medicine of University of Southern California, Los Angeles, CA 90033, USA; armstrong@usa.net; 4EO2 Concepts, San Antonio, TX 78249, USA; m.niederauer@eo2.com

**Keywords:** oxygen, hyperbaric, topical oxygen, continuous diffusion oxygen, diabetic foot ulcer, molecular biomarkers

## Abstract

Hyperbaric oxygen therapy (HBOT) and topical oxygen therapy (TOT) including continuous diffuse oxygen therapy (CDOT) are often utilized to enhance wound healing in patients with diabetic foot ulcerations. High pressure pure oxygen assists in the oxygenation of hypoxic wounds to increase perfusion. Although oxygen therapy provides wound healing benefits to some patients with diabetic foot ulcers, it is currently performed from clinical examination and imaging. Data suggest that oxygen therapy promotes wound healing via angiogenesis, the creation of new blood vessels. Molecular biomarkers relating to tissue inflammation, repair, and healing have been identified. Predictive biomarkers can be used to identify patients who will most likely benefit from this specialized treatment. In diabetic foot ulcerations, specifically, certain biomarkers have been linked to factors involving angiogenesis and inflammation, two crucial aspects of wound healing. In this review, the mechanism of how oxygen works in wound healing on a physiological basis, such as cell metabolism and growth factor signaling transduction is detailed. Additionally, observable clinical outcomes such as collagen formation, angiogenesis, respiratory burst and cell proliferation are described. The scientific evidence for the impact of oxygen on biomolecular pathways and its relationship to the outcomes in clinical research is discussed in this narrative review.

## 1. Introduction

Lower extremity complications in people with diabetes constitute a large worldwide burden within in already burdened population [1,2]. Every 1.2 s, someone with diabetes develops a foot ulcer [1]. More than half of these wounds become infected [3,4,5], leading to a high rate of emergency department visits, hospitalizations and ultimately amputation [6]. Every 20 s, someone with diabetes undergoes an amputation somewhere in the world [7,8,9]. Patients with diabetic foot ulcers are at a near three-fold greater risk for death in the year following wounding than patients with diabetes without foot ulcers [10]. This increases with additional comorbidities. Following ulceration, Charcot arthropathy, development of chronic limb threatening ischemia or amputation, 5-year mortality is comparable to most cancers [11,12,13]. Additionally, the costs for care for patients with diabetic foot ulcers exceed the cost of care most individual cancers [11,14,15]. 

The role of oxygen in wound healing has long garnered interest among researchers and clinicians alike. This interest has only increased as modalities for delivery of oxygen have evolved from large hyperbaric chambers to portable, direct topical application using localized chambers and more recently to handheld, wearable systems which continuously diffuse oxygen directly into the wound bed. There are distinct differences and advantages of each modality for oxygen delivery. Although oxygen therapy can be used for a variety of etiologies, the focus of oxygen therapy discussed revolves around its most common usage which is diabetic foot ulcerations due to its element of ischemia, either involving a macro or micro circulatory component. In hyperbaric oxygen therapy (HBOT) a contained chamber is pressurized with 100% oxygen to 2.0–2.4 atmospheres absolute for 90 min 5–7 days per week. HBOT relies on respiration and the circulatory system to deliver oxygen to the wound bed [16]. The increased pressure supersaturates the plasma; however, oxygen delivery relies on local capillary structure to reach injured tissues. Deficient or absent capillary beds may impede the oxygen delivery to ischemic tissues. Traditional TOT uses high flow oxygen concentrators coupled with chambers or bags placed directly over or around the wound. TOT applies oxygen directly to the wound, allowing the oxygen to diffuse directly into the wound and is therefore not reliant on underlying capillary structures. TOT follows an intermittent treatment regimen similar to HBOT. Currently, most recent devices are wearable and continuously generate pure, humidified oxygen from surrounding air using electrochemical oxygen generators. There is no need for an external oxygen source. They continuously diffuse oxygen (CDO) directly into wounds (24 h a day, 7 days a week) using an oxygen diffuser or oxygen diffusion dressings. CDOT, like TOT, does not depend on the underlying capillary structure of the wound bed, however, unlike TOT, the continuous application of oxygen resembles physiologic oxygen delivery. The biomolecular evidence for the effects of oxygen in wound healing including all modalities of delivery are presented. Although delivery mechanisms differ, the effect of oxygen at a cellular level is consistent. 

However, the availability of oxygen to injured tissues will depend on the method of delivery. For HBOT, which relies on inspired oxygen, the availability depends on arterial pO2, vascular supply, local capillary structures and the diffusion distance for the oxygen from the capillaries to the cells. Both edema and necrotic debris increase the diffusion distance. If the local structures are impaired or vasoconstriction is present, wound perfusion can be significantly impaired such that little to no increase in wound pO2 levels occurs despite breathing supplemental oxygen [17,18,19]. Hence, there is a need for determining local vascular adequacy using methods such as transcutaneous oxygen pressure measurement prior to initiating HBOT. Modalities that use direct application of oxygen to the wound, such as TOT and CDOT, still require adequate vascular sufficiency, yet are significantly less dependent on local capillary structures. Necrotic tissue increases the diffusion distance to the wound, so debridement is an important step to ensure optimal diffusion of oxygen into the wound bed for topically applied oxygen. Debridement has been shown to have significant benefit when applied to standard moist wound therapies [20,21,22]. The importance of debridement in topically applied oxygen was recently demonstrated in a double blind, placebo-controlled clinical study, where CDOT showed dramatically higher wound closure rates and overall closure for wounds that were debrided frequently versus those that were not [23] in patients with diabetic foot ulcers. 

The molecular processes discussed herein are oxygen dependent and do not occur without oxygen. The reactions are catalyzed by enzymes which typically have about 50% maximum speed at normal tissue pO2 levels (40–80 mm Hg) and reach 90% of maximum speed at levels varying between about 150 mm Hg to over 400 mm Hg [24,25]. These higher levels can only be achieved with supplemental oxygen. An interesting finding regarding the positive correlation between oxygen concentration and functionality is that the more oxygen there is, the faster and better the outcomes are compared to normal wound healing. The differences are even greater when compared to ischemic wounds which are hypoxic. The definitions of hypoxia and hyperoxia are relative. In the context of this review, they are relative to the levels normally found in healthy tissue surrounding a wound (40–80 mm Hg). 

## 2. Cell Metabolism and Energy

Oxygen plays a crucial role in energy production and cell metabolism. In this role, oxygen is required for intracellular processes such as biosynthesis and transport, not to mention cell survival [26]. Oxygen dependent enzymes include adenosine triphosphate (ATP) for chemical energy and nicotinamide adenine dinucleotide phosphate (NADPH) oxygenase for respiratory burst (reactive oxygen species release). ATP fuels most active cellular processes and the increased energy demand of tissue that is undergoing healing leads to a hypermetabolic state wherein additional energy is generated from oxidative metabolism [27,28,29,30]. Other metabolic processes such as aerobic glycolysis, ß-oxidation of fatty acids and the citric acid cycle are tightly attached to the energy acquisition by oxidative phosphorylation and are, therefore, oxygen dependent [31]. Conversely, when tissue oxygen levels are consistently too low (<20 mmHg pO2), cells convert to anaerobic metabolism and go into survival mode in which wound healing activities such as mitotic cell division, and, therefore, re-epithelialization with collagen production are impaired [32,33,34]. Prolonged exposure to extremely low oxygen levels, if not alleviated by oxygen, can result in cell death and tissue necrosis due to the inability of the cells to repair the spontaneous decay of cell components (DNA, RNA and proteins) and inability to maintain calcium pumps which require ATP to function [35,36]. 

## 3. Molecular Biomarkers in Growth Factor Signaling Transduction

Reactive oxygen species (ROS) are essential for the signaling processes of growth factors and processes such as leukocyte recruitment, cell motility, angiogenesis and extracellular matrix formation involved in wound healing [37]. The rate-limiting substrate for ROS production is oxygen. In a wound site, almost all wound-related cells can generate ROS using the enzyme nicotinamide adenine dinucleotide phosphate (NADPH) oxidase. The functionality of NADPH oxidase correlates positively to pO2 levels, with the maximal function of NADPH oxidase observed at pO2 > 300 mm Hg, levels only achievable with supplemental oxygen. In wounds deficient of oxygen, such as diabetic or ischemic wounds, NADPH oxidase ceases to function at pO2 levels below 20 mm Hg. There have been no noted adverse effects or increased reports of safety issues associated with high concentrations of oxygen in wound care. The increased ROS levels appear to accelerate the signaling processes without causing any damage at the cellular level. On a clinical level, studies have shown comparable or decreased adverse events and hospitalizations compared to standard of care with no supplemental oxygen [38,39,40,41].

Signal transduction of growth factors and cytokines is stimulated by ROS [42]. ROS, such as superoxide and hydrogen peroxide, increase vascular endothelial growth factor (VEGF) production in macrophages and keratinocytes [43,44]. ROS are also required for platelet-derived growth factor (PDGF) to regulate cell growth and division [45]. Like VEGF, PDGF plays a significant role in blood vessel formation (angiogenesis) [46]. ROS have effects on other processes such as cytokine action, cell motility and extracellular matrix formation [47]. Conversely, tissue hypoxia will limit redox signaling and disable the function of several growth factors such as PDGF, VEGF, keratinocyte growth factor, insulin-like growth factor one (IGF-1), transforming growth factor beta (TFG-β) and numerous molecular mechanisms (e.g., leukocyte recruitment, cell motility and integrin function) which rely on redox signaling [37,48,49]. This positive correlation between pO2 levels, ROS production, and growth factor promotion of cytokine expression explains why ischemic diabetic wounds, having little to no ROS, fail to heal and why wounds supplemented with oxygen heal faster. Typical molecular biomarkers that are indicative of wound healing are shown in Table 1 along with the processes that they are associated with. These biomarkers and their effects on wound healing will be discussed in greater detail throughout this review.

The impact of continuous diffusion of oxygen therapy (CDOT) on wound cytokines and growth factors was recently demonstrated in a prospective study of 23 patients with diabetic foot ulcers below the malleolus [50]. Results showed significant increases in growth factors, cytokines and transcutaneous oxygen pressure measurement levels after application of CDOT. Growth factors significantly increased from 280% to 820% of base levels in the first week and decreased in subsequent weeks [50] (Figure 1). Cytokines increased significantly (up to 680% compared to baseline levels) in the first two weeks and then decreased. Significant increases in transcutaneous oxygen pressure measurement indicated increased oxygen perfusion in the wound periphery. This is evidence that the topically applied oxygen not only saturated the wound bed, yet also elevated the levels of oxygen in the surrounding tissues. 

## 4. Collagen Formation

Oxygen is essential to make and properly organize collagen, which is the primary component of skin, accounting for 70–80% of dry weight and acts as the primary structural scaffold of skin and structures the matrix for angiogenesis. Organized collagen is bundled into fibers, which are interwoven and can be stretched in multiple directions without tearing. At the biomolecular level, oxygen is required for the hydroxylation of proline and lysine in procollagen [51]. Several posttranslational steps in collagen synthesis are oxygen dependent. The enzymes prolyl hydroxylase, lysyl hydroxylase and lysyl oxidase all require oxygen [46,52,53]. The formation of cross-linked triple-helices occurs via the oxygen-dependent enzyme prolyl hydroxylase and are excreted as collagen fibers. Collagen fibers are arranged into linear fibrils via cross-linking by lysyl hydroxylase. Linear fibrils are cross-linked by lysyl oxidase—a necessary step to achieve the necessary tensile strength for healed wounds. 

Higher oxygen concentrations increase the amount of collagen deposition [54] and tensile strength [55,56,57]. The rate limiting step is the rate of prolyl hydroxylation [52,53]. The oxygen level required for optimal prolyl hydryoxlase activity is at oxygen levels approaching 250 mmHg, exceeding those present in normal wounds and only achievable using oxygen therapy treatment [58,59]. It has been shown that increasing oxygen concentrations above normal physiologic levels enhances collagen synthesis and tensile strength in both animal and human subjects [55,56,57] and can increase the level of collagen organization [60]. Correction of vasoconstriction and hypoxia can result in a 10-fold increase in collagen deposition in wound repair [17,54,56,61]. The rates of collagen deposition increase as oxygen levels increase, with optimal activity at levels higher than 250 mmHg [62]. Conversely, hypoxic wounds as in patients with diabetes deposit collagen poorly and become infected easily [51,54], 

In a study using supplemental oxygen at a rate of 4 L/min through nasal cannula for 12 h a day for 3 days, it was found that three times as much collagen was deposited in patients with well-perfused and oxygenated wounds compared with those with lower oxygenation and perfusion scores [54]. A separate study using direct topical oxygen on chronic diabetic foot ulcers showed significant increases in the expression of genes associated with collagen production (TGF-β, VEGF and IL-6) during weekly follow-up visits after application of CDO in patients with diabetic foot ulcers [50].

## 5. Angiogenesis Biomarkers

The creation of new blood vessels, angiogenesis, is essential to the growth and survival of repair tissue. Oxygen levels directly affect not only the rate, yet also the quality of new blood vessel growth. Sufficient oxygen levels are required for correct collagen synthesis (posttranslational hydroxylation) [63] without which the new capillary tubes assemble poorly and remain fragile [62,64,65]. Supplemental oxygen has been shown to accelerate blood vessel growth [66]. Moderate hyperoxia increases the appearance of new blood vessels in wounds [67]. Similar to ROS activity, the rate of angiogenesis has been shown to be directly proportional to oxygen levels in damaged tissues [62], with maximum activity levels at pO2 levels exceeding 250 mm Hg.

VEGF has been shown to be a major long-term angiogenic stimulus at the wound site and is believed to be most prevalent and efficacious signal for angiogenesis. Oxygen treatment induces VEGF mRNA levels in endothelial cells and macrophages [68,69,70] and VEGF 121/165 protein expression in wounds [71]. Oxygen has also been shown to facilitate the release of VEGF165 from cell-associated stores [72]. 

Hyperbaric and topical oxygen therapy have been shown to increase VEGF expression in wounds [73] and induce angiogenesis [74]. More recently, a clinical study on gene expression of multiple factors involved in angiogenesis (VEGF, TGF-β, IL-6 and CXCL8) showed significant increases upon continuous application of oxygen (Figure 1) [50]. The expression levels over time are similar to gross observations of their effect in the field: increased redness within the first week and exudate levels that peak within the first two weeks and then subside, both indicators of new capillary formation. Furthermore, the curved response shown in Figure 1 reflects what would be expected of a chronic wound “reawakening” and entering the inflammatory stage.

## 6. Respiratory Burst Process and Cytokine Production

Oxygen is essential for respiratory burst, or the production of reactive oxygen species (ROS), used by phagocytes such as neutrophils and macrophages in bactericidal activity and the removal of necrotic cellular debris. NADPH oxidase, also known as leukocyte oxidase, has been shown to support macrophage survival, a delay of apoptosis and enables dead cell cleansing by phagocytosis [75]. NADPH oxidase in wound phagocytes, such as neutrophils and macrophages, produces superoxides (O2- and H2O2) for bactericidal activities [76]. It has been shown that up to 98% of oxygen consumed by these cells is used to produce ROS during phagocytosis [24]. Leukocyte activity, which involves the production of ROS which enables oxidative killing, is directly proportional to local oxygen concentration [77,78]. The optimal ROS production is seen at oxygen levels of greater than 300 mmHg, levels which can only be achieved with supplemental oxygen [79]. ROS activity is not restricted to phagocytes. At the wound site, ROS are generated by almost all wound-related cells [46]. The efficacy of supplemental oxygen has been shown to be similar to antibiotic administration and has additive effects when used together [80,81]. 

Interleukins are a type of cytokine protein that play important roles in the differentiation/activation of immune cells in addition to their proliferation, maturation, migration and adhesion. [https://www.ncbi.nlm.nih.gov/books/NBK499840/ StatPearls Publishing; 31 January 2021]. The addition of continuous oxygen therapy directly to a wound has been shown to increaseIL-6 significantly (up to 680% relative to baseline) in clinical studies [50,82]. IL-6 has been shown to induce chemotaxis of leukocytes into a wound [83,84]. As inflammation progresses, IL-6 signaling is responsible for the switch to a reparative environment.

## 7. Cell Proliferation Molecular Markers

Increasing oxygen levels results in faster cell proliferation, re-epithelialization and collagen formation. Fibroblast proliferation and protein production have been reported to be optimal at 160 mmHg, i.e., at pO2 levels two-fold to three-fold higher than those found in healthy tissues [85], indicating that supplemental oxygen increases the rate of wound repair. Endothelial progenitor cells (EPCs) are essential in wound healing, but their circulating and wound level numbers are decreased in diabetes. Elevated oxygen levels (hyperoxia) reverse the diabetic defect in EPC mobilization [86]. EPC mobilization into circulation is triggered by hyperoxia through induction of nitric oxide with resulting enhancement in ischemic limb perfusion and diabetic wound healing [87,88,89]. 

Matrix metalloproteinases (MMPs) are a group of enzymes responsible for degrading a majority of extracellular matrix proteins during tissue development, growth and turnover [90,91]. MMPs have diagnostic, predictive and indicative power for wound healing and can be measured from wound fluid. They are required for a wound to heal properly, at a suitable level, in the correct position and for a certain length of time. Excess activity may lead to a chronic non healing wound. Chronically increased levels of MMPs and reduced levels of TIMPs (MMP regulators), or just abnormalities in their ratio, are associated with non-healing. Studies show that medical interventions which aid in lowering MMP activity will promote the healing of stalled wounds and that decreasing MMP-2 tissue levels will result in wound healing. In one study elevated MMP-1 and TIMP-1 levels were noted in on oxygen treatment group, yet not in the control group [82].

At the clinical level, the cumulative effects of oxygen in all the various aspects discussed herein result in significant real-world results. In a clinical study which analyzed VEGF expression versus wound size reduction using TOT, a significant correlation between wound closure and VEGF expression was found [73]. Recent results using topically applied oxygen therapy, both continuously and intermittently, on diabetic foot wounds has been shown to increase the rate of wound closure, by as much as 460% relative to moist wound therapy in several double-blinded trials, two of which had placebo control groups [23,38,39,40,41]. As would be predicted by the positive correlation of various mechanisms of action to the relative concentration of oxygen, wounds that were larger, deeper, more chronic and weight-bearing had improved responses relative to controls than those that were smaller, shallower, less chronic or non-weight-bearing, respectively [38]. 

## 8. Summary

Biomolecular pathways associated with wound care have been shown to be positively correlated to local tissue oxygen concentration. Maximal activity levels of the related enzymes, growth factors and cytokines, as well as the associated physiological processes, are significantly above the levels normally found in healthy tissues. The positive, differential effect from supplemental oxygen has been shown to be even higher for tissues with compromised blood supply leading to ischemic diabetic wounds. Increasing the levels of oxygen in afflicted tissues significantly increases not only the rate, yet also the quality of tissue repair. These elevated levels of oxygen can only be achieved through supplemental oxygen therapy, whether it be respiratory based (HBOT) or directly applied to the wound (TOT, CDOT), all of which are reliant on diffusion gradients. The recent research on the scientific basis and clinical outcomes of oxygen therapy lays a foundation for further research in molecular biomarkers utilizing oxygen therapy.

## Figures and Tables

**Figure 1 biomolecules-11-00925-f001:**
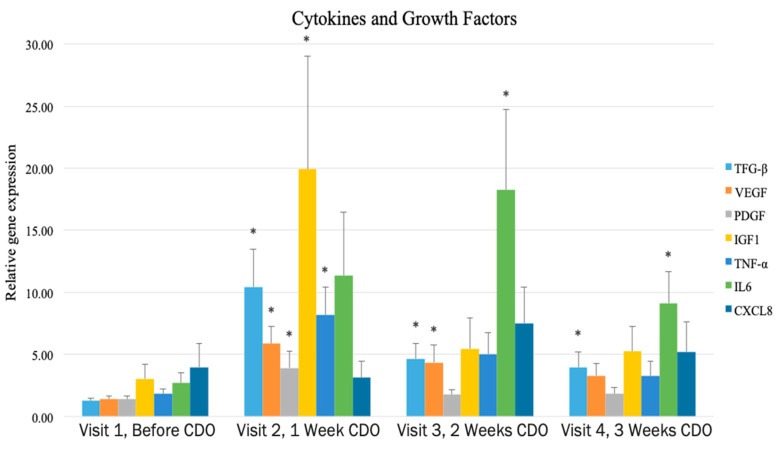
Impact of CDO on gene expression for various cytokines and growth factors at each visit Asterisks indicate significant increase from baseline [50]. Adapted with permission.

**Table 1 biomolecules-11-00925-t001:** Molecular Biomarkers and Clinical Impact in Wound Healing.

Growth Factors	
IGF-1	protein production and cell proliferation and migration
PDGF	cell growth and division and chemotaxis
TGF-β	angiogenesis, fibroblast proliferation, collagen synthesis and deposition, extracellular matrix (ECM) remodeling, tissue remodeling, granulation tissue stimulant and anti-inflammatory mediator
VEGF	angiogenesis and collagen deposition and epithelialization
IGF-1	protein production and cell proliferation and migration
cytokines	
CXCL8	angiogenesis, epithelialization, fibroblast migration and inflammatory mediator
IL-6	leukocyte infiltration, angiogenesis, collagen accumulation, anti-inflammatory, granulation tissue stimulant and mitogenic
TNF-α	leukocyte recruitment, cell regulator, ECM synthesis and inflammatory mediator

## Data Availability

Data available on PubMed.

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
