# Peer review of "Molecular Biomarkers of Oxygen Therapy in Patients with Diabetic Foot Ulcers"

_biomolecules, 2021, doi:10.3390/biom11070925_

Round 1

Reviewer 1 Report

In this commentary, the authors summarized the current understanding of oxygen therapy for diabetic foot ulcers, focusing on molecular biomarkers. I believe that the present manuscript has a comprehensive, interesting and current approach and could be a starting point for a broader analysis for a better understanding of the impact of this therapy on biomolecular pathways related to wound healing. My overall impression is that this good commentary would be of great interest to researchers working in the field of diabetes complications and beyond.

I only have three comments:

  • The introduction would need more information on the diabetic foot and its etiopathogenesis.
  • A table is needed that summarizes the different molecular biomarkers for diagnosis and outcome in diabetic foot ulcer.
  • Has any adverse effect of the use of this therapy been contemplated? If so, it might help to write a section on this assessment, especially related to excessive ROS production or whether there is a consequence on the functionality of the mitochondria.

Author Response

  • The introduction would need more information on the diabetic foot and its etiopathogenesis.  Additional information was added to the introduction.
  • A table is needed that summarizes the different molecular biomarkers for diagnosis and outcome in diabetic foot ulcer.  A table was included.
  • Has any adverse effect of the use of this therapy been contemplated? If so, it might help to write a section on this assessment, especially related to excessive ROS production or whether there is a consequence on the functionality of the mitochondria.  A brief section was added.

Reviewer 2 Report

This is a very interesting paper, which I´ve enjoyed reading it. However there´s some minor changes that you should made in order to improve it.

Is it a narrative review ? Please state that in the title

Which are the clinical implications ? Could you add a little paragraph about it. 

Line 265. Author Contributions. In that section you must to expose the exact contributions of each author

Line 272. Please, remove the first part of the sentence "Declare conflicts of interest or state"

Author Response

This is a very interesting paper, which I´ve enjoyed reading it. However there´s some minor changes that you should made in order to improve it.

Is it a narrative review ? Please state that in the title.  This paper is a commentary and will be stated in the publication.

Which are the clinical implications ? Could you add a little paragraph about it.  A paragraph regarding the clinical implications was added. 

Line 265. Author Contributions. In that section you must to expose the exact contributions of each author.  The exact contributions were exposed.

Line 272. Please, remove the first part of the sentence "Declare conflicts of interest or state".  "Declare conflicts of inferest or state" was removed.